# *Dendrobium officinale* Polysaccharides Inhibit 1-Methyl-2-Nitro-1-Nitrosoguanidine Induced Precancerous Lesions of Gastric Cancer in Rats through Regulating Wnt/β-Catenin Pathway and Altering Serum Endogenous Metabolites

**DOI:** 10.3390/molecules24142660

**Published:** 2019-07-23

**Authors:** Yi Zhao, Bingtao Li, Gaoyu Wang, Shuchao Ge, Ximing Lan, Guoliang Xu, Hongning Liu

**Affiliations:** 1Research Center for Differentiation and Development of Basic Theory of Traditional Chinese Medicine, Jiangxi University of Traditional Chinese Medicine, Nanchang 330004, China; 2Jiangxi Province Key Laboratory of TCM Etiopathogenisis, Nanchang 330004, China

**Keywords:** precancerous lesions of gastric cancer (PLGC), *Dendrobium officinale* polysaccharides (DOP), Wnt/β-catenin pathway, serum endogenous metabolites analysis

## Abstract

*Dendrobium officinale* is a herb in traditional Chinese medicine where *D. officinale* polysaccharides (DOP) are the main active ingredient. This study aimed at evaluating DOP efficiency at inhibiting 1-Methyl-2-nitro-1-nitrosoguanidine (MNNG) induced precancerous lesions of gastric cancer (PLGC) in rats through the Wnt/b-catenin pathway and analyzing the variations of serum endogenous metabolites. PLGC was established in male Sprague-Dawley (SD) rats by administering 150 μg/mL MNNG in drinking water for 7 months and giving 0.1 mL of 10% NaCl once weekly during the initial 20 weeks. Treatment with DOP inhibited the progress of PLGC through decreasing the expression of β-catenin by immunohistochemical analysis. The futher study indicated DOP downregulated gene expression of Wnt2β, Gsk3β, PCNA, CyclinD1, and β-catenin, as well as protein expression of Wnt2β, PCNA, and β-catenin. On the other hand, there were nine endogenous metabolites identified after the DOP treatment. Among these, the most significant one is betaine because of its strong antioxidant activity, leading to an anti-tumor effect. DOP can inhibit MNNG-induced PLGC models via regulating Wnt/β-catenin pathway and by changing endogenous metabolites.

## 1. Introduction

Gastric cancer is a common malignant digestive tract tumor, with the latest report showing that its incidence at 5.7% and its mortality at 8.2% in the world [1]. Precancerous lesions of gastric cancer (PLGC) is the most important phase in the progression of gastric cancer, in that it presents the best opportunity for inhibiting tumor progression [2,3]. The Wnt pathway is one of the most important signaling pathways involved in gastric carcinogenesis. It is well documented that the Wnt pathway is constitutively activated and beta-catenin accumulation is changed in the progression of the gastric tumorigenesis. Therefore, the Wnt pathway is recognized as a useful pathway to examine the relationship between gastric tumorigenesis and drug effects, as well as to explain the mechanism of its preventive effect [4,5].

A metabolomics approach is carried out to estimate the effects of treatment and differences among groups and subsequently to know the cause of such differences. The differences are mainly presented by the changes of endogenous small metabolite molecules, which are characterized metabolic pathways of biological systems. It is usually combined with pattern recognition technology, such as structural mass spectrometry (MS) [6,7]. At present, metabolomics is commonly used in pathology, pharmacology, and toxicology, and it is helpful for finding biomarkers and exploring metabolic networks of the drug [8]. Partial least-squares-discriminant analysis (PLS-DA) is a popular multivariate statistical analysis tool, which can be applied to these complex spectral data to express the characterization of changes related to the drug effect or disease [9].

*Dendrobium officinale Kimura & Migo* (*D. officinale*) (“Tie pi Shi hu”) is a useful traditional Chinese medicine (TCM) believed, for thousands of years, to reinforce stomach function and generate body fluid. The National Health Commission of the People’s Republic of China characterizes it as a herb and a health food. We also conducted the urine metabolomics analysis of *D. officinale* extraction on gastric precancerous lesion [10,11], and the result indicated melatonin was one of the important endogenous metabolites related to the growth inhibition of gastric cancer [12]. *D. officinale* polysaccharides (DOP) is the main effective component in *D. officinale*. Our previous studies showed that DOP could inhibit the progression of gastric carcinogenesis, and the suppression are associated with the reduction of 8-OHdG levels, as well as the activation of the NRF2 pathway and its related antioxidant enzymes, HO-1 and NQO-1 [13]. However, the process of its inhibitive effects through regulating the Wnt/β-catenin pathway and endogenous metabolites remain unclear. In this study, we further explored the mechanism of DOP inhibition on PLGC progress focusing on the activation of the Wnt/β-catenin pathway and the alteration of endogenous metabolites in serum.

## 2. Result

### 2.1. Immunohistochemical Analysis of β-Catenin

The protein expression of β-catenin was significantly increased in the PLGC group and the average density was significantly higher than that of the normal group (Figure 1B,F). After DOP administration, the high and medium doses could notably decrease the expression of β-catenin and both had a significant difference compared to the CTRL group (all *p* < 0.05) (Figure 1C,D,F).

### 2.2. RT-PCR Analysis of Gene related to Wnt/β-Catenin Pathway

In the current experiment, RT-PCR analysis showed the expression of Wnt2β, Gsk3β, PCNA, and β-catenin mRNA strongly increased in the PLGC model (all *p* < 0.05). After treatment with all the different doses of DOP, the four genes were all significantly decreased and had the statistical differences when compared with the PLGC model group, but the expression of CyclinD1 only had a downward trend (Figure 2).

### 2.3. Western Blot Analysis of Wnt2β, PCNA, and β-Catenin Protein Expression

As shown in Figure 3A–D, the protein expression of Wnt2β, PCNA, and β-catenin were significantly increased in the PLGC group compared to the CTRL group (all *p* < 0.05), and these expression levels were attenuated by three doses of DOP, especially in PCNA and β-catenin expression. But CyclinD1 protein expression had no remarkable change (Figure 3E).

### 2.4. Metabolites Profiles in the Serum

According to the PLS-DA score plot in Figure 4A,B, the analyzed result shown that R2X = 0.768, R2Y = 0.781, Q2 = 0.653 in the negative (Figure 4A) and R2X = 0.541, R2Y = 0.864, Q2 = 0.817 in the positive model (Figure 4B), that means the PLS-DA score plot provided an intuitive summary of the sample clustering patterns. The different circle in the figure indicated that the PLGC model group were gathered together and distributed differently from the CTRL group, which means the metabolic profiles of PLGC rats were far from those of CTRL group. After treatment with DOP, the data show that R2X = 0.727, R2Y = 0.782, and Q2 = 0.608 are in the negative, while R2X = 0.713, R2Y = 0.772, and Q2 = 0.637 are in the positive model; all the R2 values were high which indicated a high predictive accuracy. The results indicated that the DOP treated group were far from the model group, and demonstrated that the PLS-DA analysis was the optimal discriminant mode for these groups.

### 2.5. Potential Endogenous Metabolites Identification

The VIP value and Student’s t-test of the p-value were used to reflect the importance of the metabolites. Variables with VIP > 1.0 and *p* < 0.05 were selected as potential biomarkers for further statistical analysis. Nine endogenous metabolites were identified, they were Betaine, 5′-Deoxyadenosine, Enterolactone, Metanephrine, Proline betaine, Imidazolelactic acid, l-leucyl-l-proline, 15,16-DiHODE, Alpha-CEHC (Table 1), then compared the value of highlight between different groups (Figure 5).

### 2.6. Determination of Endogenous Metabolites

The secondary mass spectrometry was identified by the HMDB and METLIN databases. Taking the metabolite of *m*/*z* 252.1095 in a positive ion mode as an example, the *m*/*z* value of second mass spectrum is 136.0612 (Figure 6A), which matches the database, and then the substance is determined to be *m*/*z* 252.1095 (Figure 6B). Another eight metabolites were determined in the same way.

### 2.7. Enrichment Analysis of Differential Endogenous Metabolites

Using MetaboAnalyst 3.0 for metabolite enrichment analysis, metabolites were mainly concentrated in the four pathways, they were betaine metabolism, methionine metabolism, glycine, serine, and threonine metabolism, and tyrosine metabolism. Among them, betaine metabolism is the most significant pathway (Table 2, Figure 7).

## 3. Discussion

The Wnt signaling pathway is implicated in many hereditary diseases and tumorigenesis, including colorectal cancer, hepatocellular carcinoma, and gastric cancer [14]. It is reported that *Helicobacter pylori* could activate oncogenic c-Met and epidermal growth factor receptor (EGFR), inhibit tumor suppressor Runx3 and Trefoil factor 1 (TFF1), then stimulate the Wnt/β-catenin pathway [15]. The Wnt/β-catenin pathway also has a high expression in the MUN-induced gastric cancer mice model. WNT2 is one of the proto-oncogenes with the potential to activate the WNT-β-catenin-TCF signaling pathway; it promotes cell migration and invasion on the progression of gastric cancer [16]. The accumulation of β-catenin in the nucleus could promote the transcription of many oncogenes, such as c-Myc and CyclinD-1, and it is also found in the MNNG-induced glandular stomach adenocarcinomas [17]. Cyclin D1 is overexpression in MNNG-induced rat gastric adenocarcinomas [18]. In our experiment, the gene and protein expression of Wnt2β, β-catenin, and PCNA were strongly enhanced in the PLGC group, this result matched the report. After treatment with DOP, all of them were declined and had statistical differences compared to the PLGC group. The result showed the DOP could inactivate the Wnt/β-catenin pathway to inhibit the progression of gastric cancer in rats.

Metabolomics has been widely used in tumors and was mainly focused on the discovery of tumor biomarkers, clinical diagnosis of tumors, and evaluation of treatment methods [19]. In gastric cancer, a large number of related studies on metabolomics have been carried out. Corona [20] investigated serum metabolomic profiles to find additional biomarkers that could be integrated with serum PGs and G17 to improve the diagnosis of GC and the selection of first-degree relatives (FDR) at a higher risk of GC development; the result showed that the predictive risk algorithm composed of the C16, SM (OH) 22:1 and PG-II serum levels could be used to stratify FDR at a high risk of GC development. Gu [21] performed an NMR-based metabolomics analysis of a rat model of gastric carcinogenesis, the result showed two activated pathways (glycolysis; glycine, serine, and threonine metabolism) substantially contributed to the metabolic alterations at the GC different stage. In our results, after using MetaboAnalyst 3.0 for metabolite enrichment analysis, the metabolites were mainly concentrated in four pathways: betaine metabolism, methionine metabolism; glycine, serine and threonine metabolism, and tyrosine metabolism. All of which are in agreement with existing research.

In these nine endogenous metabolites, the most interesting one is betaine, which is an important methyl group donor, participating in methionine recycling and phosphatidylcholine synthesis [22]. Recently the report showed that betaine can alleviate inflammation by lowering interleukin (IL)-1beta secretion and improve intestinal function by enhancing digestive enzymes, ameliorating intestinal morphology, and enriching intestinal microbiota of high-salt stressed rats [23,24]. In our research, after treatment with DOP, medium and low doses of DOP significantly increased the level of betaine, which was statistically significant compared with the model group (*p* < 0.01). We found that the PLGC model rats had a side effect on liver function which increased the ALT level (*p* < 0.001); high, medium, and low doses of DOP significantly decreased the level of ALT, which was statistically significant compared with the model group (*p* < 0.01). The results indicate that DOP has a protective effect on liver function [13], which may be related with the upregulation of betaine by DOP, although further research is required. In another way, the report indicated betaine could attenuate DEN-induced hepatocarcinogenesis through the down-regulation of *P*16 and inhibiting the up-regulation of c-Myc, which cued the betaine could has the inhibition on gastric carcinogenesis [25].

Alpha-CEHC (α-CEHC) is a water-soluble metabolite of vitamin E, which circulates in the blood and is excreted with the urine. The report had concluded that alpha-CEHC is a molecule with good antioxidant activity [26]. Alpha- and γ-tocopherol, as well as their final metabolites α- and γ-CEHC, suppressed cyclin D1 expression and inhibited PC-3 prostate cancer cell proliferation [27]. In our experiment, the PLGC group can downregulate the expression of α-CEHC and after administering DOP, it upregulated, this result indicated the strong anti-oxidant capacity of DOP, and it was the same as the report [28]. In our previous study, we had certified that DOP could promote NRF2 into the nucleus and increase downstream anti-oxidant enzyme capacity [13].

Enterolactone is a bioactive phenolic metabolite known as a mammalian lignan derived from dietary lignans. It has been reported that enterolactone exhibited anti-cancer properties against breast cancer [29], ovarian cancer [30], prostate cancer [31], and lung cancer [32]. It also had a clear role in preventing cancer progression at different stages, reducing risk, decreasing the mortality rate and improving overall survival [33]. The data showed that the height value of Enterolactone was decreased in the PLGC model, and increased after administering DOP, this result indicated the inhibition of DOP on the PLGC model maybe due to its prevention of cancer progression.

Other endogenous metabolites that have no relationship with cancer, gastric cancer, and polysaccharides, were imidazolelactic acid, 5′-Deoxyadenosine, Methoxytyramine, Proline betaine, l-leucyl-l-proline. Imidazolelactic acid is the component of normal human urine, histidine loading causes an increase in the excretion of imidazolelactic acid. Urinary excretion of imidazolelactic acid is also an indication for folic acid and vitamin B12 deficiency. Histidine deficiency attenuates cell viability in rat intestinal epithelial cells by apoptosis via mitochondrial dysfunction [34]. In our experiment, the level of imidazolelactic acid was increased in the PLGC group, which was probably related to the high level of histidine. The 5′-Deoxyadenosine (5′-dAdo), which is a substrate for 5′-methylthioadenosine phosphorylase, was rapidly cleaved to adenine by cell-free, had been shown to have a number of biochemical effects on Ehrlich ascites cells [35]. Methoxytyramine, one of the O-methylated metabolites of catecholamines, was used to discriminate different hereditary forms of pheochromocytoma, which indicated that patients with MEN2 and NF1 genes presented with tumors characterized by increased plasma concentrations of metanephrine [36,37]. Proline betaine is known to act as an osmoprotectant in citrus fruit, alfalfa sprouts, molluscs, and bacteria. Proline betaine concentrations were reported to be higher in plasma and urine after orange juice consumption so it was regarded as a putative biomarker of citrus consumption [38]. l-leucyl-l-proline, also named Leucylproline, is a proteolytic breakdown product of larger proteins. It belongs to the family of Peptides [39]. 15,16-DiHODE is an oxygenated lipid found in human blood, which belongs to the main class of octadecanoids and the sub class of other octadecanoids [40]. These metabolites had no relationship with our experiment.

## 4. Material and Methods

### 4.1. Animals

Male Wistar rat (70 to 90 g) were purchased from Beijing Vital River Laboratory Animal Technology Co., Ltd. (Certificate No. SCXK 2015-0002, Beijing, China). The rats were fed in the standard experimental conditions (room temperature 23 ± 1 °C, relative humidity 55 ± 5%) with a 12 h light/dark cycle and received food and water. Before the experiment, the rats were adapted to the experimental environment for 1 week. The protocol for these experiments was approved by international ethical guidelines and the Institutional Animal Care and the Animal Ethics Committee of Jiangxi University of Traditional Chinese Medicine.

### 4.2. Drugs and Reagents

*Dendrobium officinale* extraction (DOE) was purchased from Zhejiang Shou Xian Valley Medical Limited by Share Ltd. (Jinhua, Zhejiang, China), the fresh *D. officinale* added 20 fold volume of water and extracted 2 h one time, after filtration then evaporated. This process was repeated three times. In the end, formed the water extraction (DOE) [41]. The DOE was added 5 times 95% alcohol and mixed, after keeping 24 h, the mixture was centrifuged at 4000 rpm for 15 min and the supernatant was removed. This process was repeated twice. The precipitate was then washed sequentially with ethanol and acetone then fully dissolved with 80% ethanol. In the end, the precipitate was dried to dry powder and formed crude DOP. The polysaccharide content is 83%, which was determined by phenol-sulfuric acid; the molecular weight, which was detected by High-Performance Gel-Permeation Chromatography (HPGPC), was 3500 and 1,000,000. This protocol of preparation and content of DOP was reported before [13]. The gel chromatography column used (TSK-gel, G4000PWXL, 7.8 mm × 300 mm) was obtained from TOSOH (TOSOH, Tokyo, Japan). Total RNA Extraction Kit and Reverse Transcription PCR kits were the products of the Promega Corporation (Promega, WI, USA). Fast Start Essential DNA Green Master was the product of Roche (Roche, Basel, Switzerland). Anti-Wnt2βantibody, Anti-CyclinD1, Anti-β-Catenin, Anti-PCNA antibody, and Anti-β-Actin were all purchased from Abcam Company (Abcam, Cambridge, Britain).

### 4.3. Inhibition Effects of DOP on PLGC in Rats

Sixty male rats were randomly divided into five groups: A, the CTRL group (the normal rats were administered with free water and diet); B, the PLGC model group (PLGC) (The rats were given 150 μg/mL MNNG in drinking water for 7 months and given 0.1 mL of 10% NaCl once weekly during the initial 20 weeks, which regarded as PLGC model [42,43]); C–E, the PLGC model were administered with 9.6, 4.8, 2.4 g/kg mg/kg/day DOP, respectively (L-DOP, M-DOP and H-DOP), and they were administered DOP 2 weeks earlier before starting the PLGC model (Figure 8). The body weight of the rats were recorded every week. After seven months, all animals were sacrificed by intraperitoneal injection of Thiopental. The serum from the inferior vena cava was collected in a tube and centrifuged at 3000 rpm at 4 °C for 10 min. The stomach was opened along the greater curvature on ice, then divided into three parts for histological study, RT-PCR, and Western Blot Analysis.

### 4.4. Immunohistochemistry

4-μm thick sections were dewaxed with xylene and hydrated using sequential ethanol (100, 95, 85, and 75%) and distilled water. Antigen retrieval was performed by heating sections in 0.01 M sodium citrate buffer (pH 6.0). Tissue slides were incubated overnight with β-catenin antibody (dilution 1:1000) at 4 °C, and then immunostained with secondary antibodies. Sections were incubated with 3, 3′-diaminobenzidine (DAB) to produce a brown product and counterstained with hematoxylin. The positive staining was evaluated. Five fields of view were randomly selected for each slice under a 400-fold microscope (Nikon, Tokyo, Japan). Image Pro Plus 6.0 software (Media Cybernetics, Rockville, MD, USA) was used to mean density analysis.

### 4.5. Total RNA Extraction and Quantitative Real-Time RT-PCR

The total RNA samples were obtained from stomach tissues using Eastep^®^ Super Total RNA Extraction Kit (Promege) following the manufacturer’s protocol. The purity of the extracted RNA was quantified by Nanodrop 2000 (Thermo Scientific, Waltham, MA, USA), and then the RNA samples were transcribed into cDNA using GoScript™ Reverse Transcription System (Promege) following the manufacturer’s instructions. The sequences of the primers (Sangon Biotech, Shanghai, China) are shown in Appendix A. β-actin was used as a reference standard. Subsequently, RT-P CR were done by using Fast Start Essential DNA Green Master (Roche, Switzerland). The conditions for RT-PCR were 95 °C (10 min; preheating), 40 cycles of thermal cycle (95 °C, 10 s; 55 °C–60 °C, 30 s; 72 °C, 10 s), melt curve (95 °C, 10 s; 65 °C, 60 s). The relative expression for a particular gene was calculated using the Ct method, a comparative 2^−ΔΔCT^ method. The mean value of individual animals was calculated and changed in the number of folds in comparison to veh/veh was noted.

### 4.6. Western Blot Analysis

The total protein samples from stomach tissues were extracted following standard protocols according to the manufacturer’s protocol (Beyotime Biotechnology, shanghai, China) and the protein content was determined using the BCA protein assay kit (cwbiotech, Beijing, China). Proteins were subjected to SDS-PAGE (10–15%) and then were transferred to PVDF membranes (Millipore, Burlington, MA, USA). After blocking nonspecific binding sites with 5% dried skim milk, the membranes were incubated overnight at 4 °C with primary antibodies. The blots were incubated with horseradish peroxidase-conjugated antibodies for 2 h at room temperature and detection and imaging were performed using an enhanced chemiluminescence system and a ChemiDoc™ XRS Imaging System (Bio-Rad Laboratories, Hercules, CA, USA). Intensity values expressed as the relative protein expression were normalized to β-actin.

### 4.7. Preparation of Plasm and UPLC/Q-TOF-MS Analysis

200 μL serum was mixed with 800 μL methanol and acetonitrile with the ratio of 1:9 and vortexed 2 min, then placed 3 h in 4 °C. After centrifugation at 13,000 rpm in 4 °C for 10 min, the supernatant was dried with nitrogen and reconstituted in 400 μL 15% methanol, vortexed 1 min again and centrifugation at 13,000 rpm in 4 °C for 10 min. Chromatographic separation was performed on an Acquity BEH C18 UPLC column (1.7 μM, 2.1 × 100 mm; Waters, Milford, MA, USA) at 40 °C with a flowrate of 0.3 mL·min^−1^, and the sample chamber temperature was 10 °C. The mobile phase was composed of acetonitrile (A) and water containing 0.1 % formic acid (Sigma-Aldrich, MO, USA) (B), The optimized UPLC elution conditions were 1–4.5 min, 1–56% B; 4.5–10 min, 56–100% B; 10–12 min, 100% B; 12–20 min, 100–90%, 20–22 min, 90–1%. The injection volume was 1 μL.

The mass spectra were acquired in both negative and positive modes by using A SYNAPY G2 high-definition mass spectrometer (Waters, Manchester, USA), and under the following conditions: capillary voltage, 2.8–3.0 kV; sample cone voltage, 40 V; extraction cone voltage, 80 V; scan time was set to 0–13 min; data were collected from *m*/*z* 50–1000 Da. The nitrogen gas desolation rate was set to 800 L/h at 400 °C; the cone gas rate was 50 L/h; and the source temperature was 100 °C. In order to ensure accuracy and repeatability, the standard curve was established by using sodium formate, Leucine-enkephalin (Waters, Manchester, USA) was used as the lock mass generating [M + H]^+^ (*m*/*z* 556.2771) to ensure accuracy during MS analysis and dynamic range enhancement was applied. The tandem mass spectrometer with a low collision energy was 4 eV and a high collision energy was 20 to 40 eV. Data acquisition and processing were performed using Mass Lynx v.4.1 (Waters, Milford, MA, USA).

### 4.8. Metabolomics Data Analysis

The data extracted from the metabolite information includes retention time, exact mass and abundance, and then was imported into QI software for data processing. The ion peaks were aligned using retention time and accurate mass, and the retention time deviation was ±0.1 min and the mass deviation was ±5 ppm. After the component identification was completed, the identified data was imported into the Ezinfo software for partial least squares-discriminant analysis (PLS-DA) analysis, and the components with VIP greater than 1.0 were filtered out. Endogenous metabolites were initially identified from the Human Metabolome Database. The METLIN Metabolomics Database was used to analyze the secondary fragment map of the component. Student’s t-test was used for SPSS 20.0 (Chicago, IL, USA) to evaluate the significant differences of potential biomarkers. The MetaboAnalyst 3.0 (http://www.metaboanalyst.ca/) database was used for metabolic pathway analysis.

## 5. Conclusions

The present results show that DOP could effectively inhibit the Wnt/β-catenin pathway. Meanwhile, DOP could notably regulate the nine endogenous metabolites in the PLGC rat model and the most significant one is betaine because of its strong antioxidant activity, which leads to an anti-tumor effect.

## Figures and Tables

**Figure 1 molecules-24-02660-f001:**
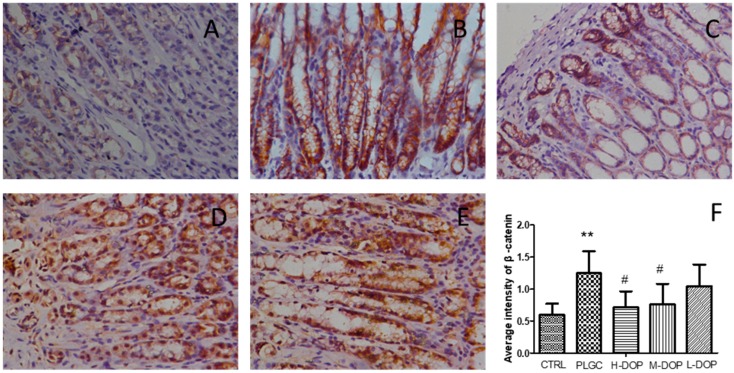
The protein expression of β-catenin in stomach tissues by immunohistochemistry (400 × original magnification). (**A**): The CTRL group: the normal rat were administered with free water and diet; (**B**): The rat were given 150 μg/mL 1-Methyl-2-nitro-1-nitrosoguanidine (MNNG) in drinking water for 7 months and given 0.1 mL of 10% NaCl once weekly during the initial 20 weeks, which was regarded as the PLGC model; (**C**–**E**): The PLGC model were administered with 9.6, 4.8, 2.4 g/kg/day *Dendrobium officinale* polysaccharide (DOP), respectively, and they were administered DOP 2 weeks earlier before starting the PLGC model. (**F**): The intensity of β-catenin measured by Image J. Data was presented as mean ± SEM (n = 6). Statistical analyses were carried out by using one-way ANOVA and Tukey post-hoc test. ** *p* < 0.01 vs. CTRL; ^#^
*p* < 0.05 vs. PLGC model.

**Figure 2 molecules-24-02660-f002:**
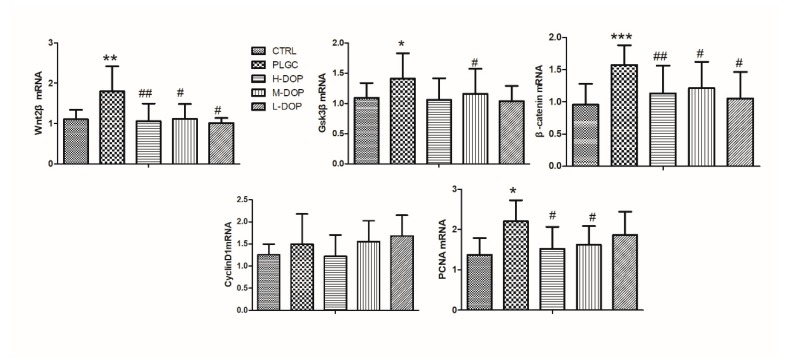
The effect of DOP on Wnt2β, Gsk3β, PCNA, and β-catenin gene expression. Results from administering DOP 7 months on PLGC rat model. The relative expression for a particular gene was calculated by using Ct method, a comparative 2^−ΔΔCT^ method. Data was presented as mean ± SEM (n = 6). Statistical analyses were carried out by using one-way ANOVA and Tukey post-hoc test. * *p* < 0.05, ** *p* < 0.01 and *** *p* < 0.001 vs. CTRL; ^#^
*p* < 0.05 and ^##^
*p* < 0.01 vs. PLGC model.

**Figure 3 molecules-24-02660-f003:**
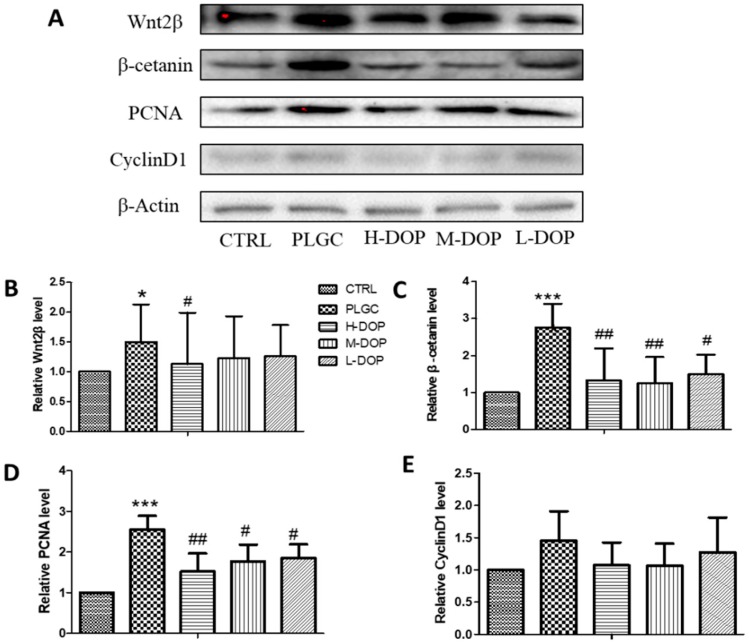
The effect of DOP on Wnt2β, PCNA and β-catenin protein expression ((**A**) the western blot bands; (**B**–**E**) the relative level of Wnt2β, PCNA, β-catenin and CyclinD1). Results from administering DOP 7 months on PLGC rat model. The relative expression for protein was analyzed and calculated by using Image lab software. Data was presented as mean ± SEM (n = 3). Statistical analyses were carried out by using one-way ANOVA and Tukey post-hoc test. * *p* < 0.05, ** *p* < 0.01 and *** *p* < 0.001 vs. CTRL; ^#^
*p* < 0.05 and ^##^
*p* < 0.01 vs. PLGC model.

**Figure 4 molecules-24-02660-f004:**
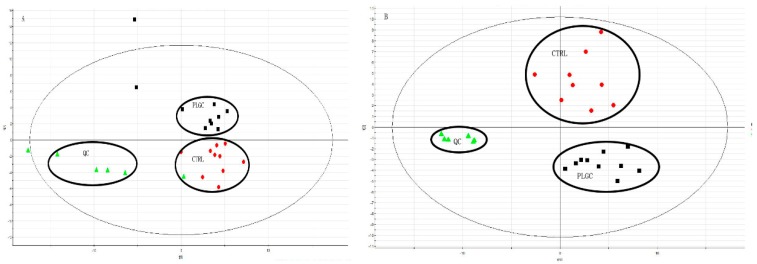
PLS-DA analysis between different groups in positive and negative ion pattern. After the component identification was completed, the identified data was imported into the Ezinfo software for partial least squares-discriminant analysis (PLS-DA) analysis, the circle showed that they tend to be grouped together in the PLS-DA analysis, and could be separated between each groups of observations, then cited there have the change in PLGC group (**A**,**B**). After treatment with DOP (**C**,**D**), the gather was changed, which indicated that the endogenous metabolites could be different from the PLGC group.

**Figure 5 molecules-24-02660-f005:**
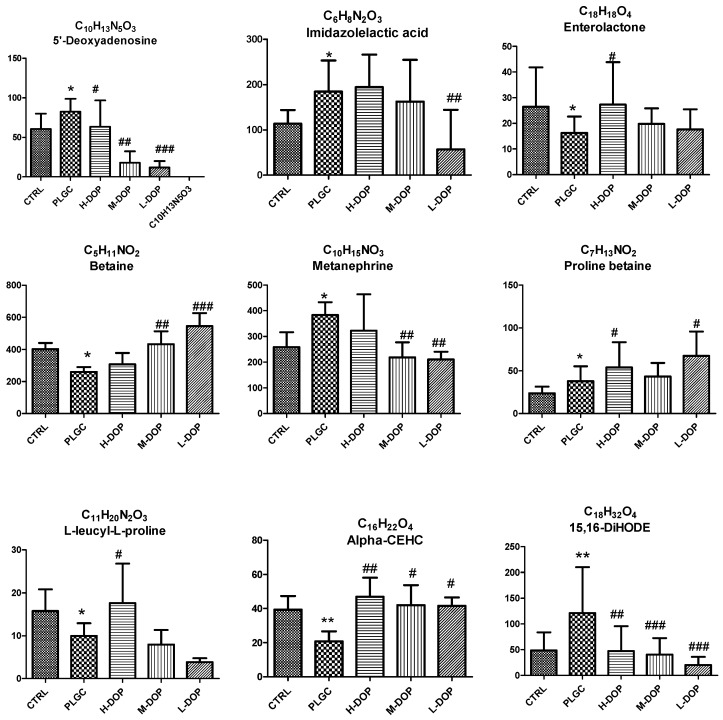
The effect of DOP on height values of nine changing endogenous metabolites in each group under positive and negative ion mode. Results from administering DOP 7 months on PLGC rat model: the compound was analyzed in the serum by UPLC/Q-TOF-MS and then selected variables with a VIP > 1.0 and *p* < 0.05 and that compound was regarded as potential biomarkers for further statistical analysis. Statistical analyses were carried out by using one-way ANOVA and Tukey post-hoc test. * *p* < 0.05, *** *p* < 0.001 vs. the CTRL; ^#^
*p* < 0.05, ^##^
*p* < 0.01 and ^###^
*p* < 0.001 vs. the PLGC model.

**Figure 6 molecules-24-02660-f006:**
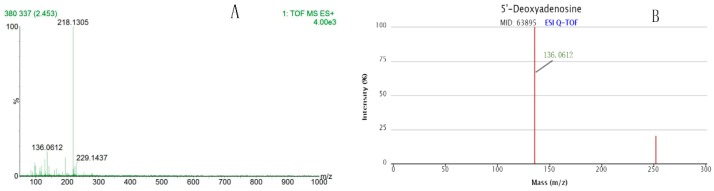
To confirm the compound by two stages mass spectrometry analysis. Taking the metabolite *m*/*z* 252.1095 as an example, the first stage of mass spectrometry (MS1) is 252.1095 and second stage of mass spectrometry (MS2) is 136.0612 (**A**), next to compare with the METLIN metabolomics database, the MS2 136.0612 was also appeared (**B**), and then confirmed the MS1 of this compound was 252.1095. Student’s t-test was used for statistical analysis to evaluate significant differences of endogenous metabolites. (**A**): Two stages mass spectrometry of the sample; (**B**): Two stages mass spectrometry from the database.

**Figure 7 molecules-24-02660-f007:**
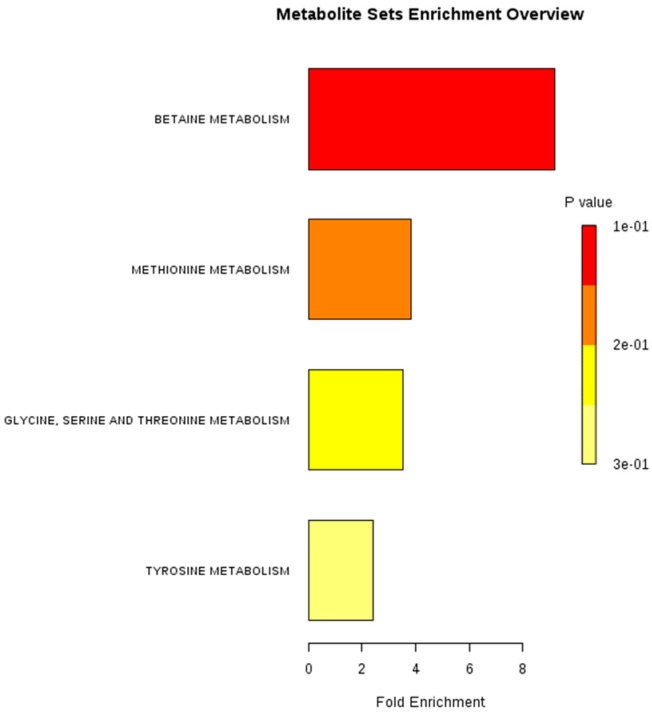
The effect of DOP on the metabolites pathway. By uploading the nine endogenous metabolites to the MetaboAnalyst software, metabolite enrichment analysis could be conducted.

**Figure 8 molecules-24-02660-f008:**
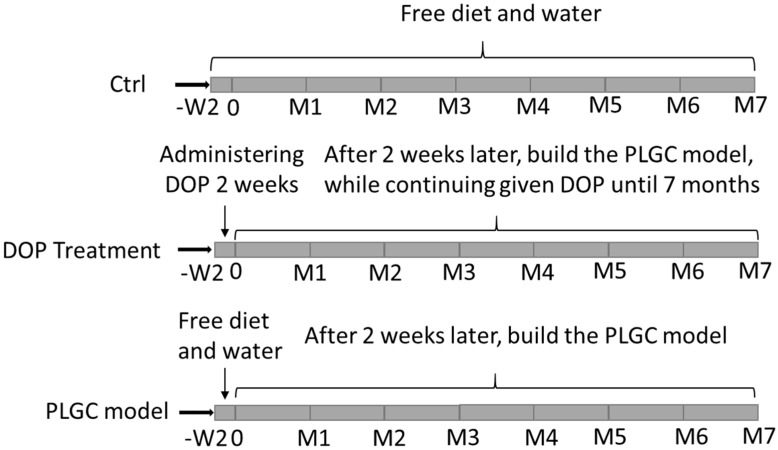
The diagrammatic sketch of the experiment.

**Table 1 molecules-24-02660-t001:** Preliminary identification results of endogenous metabolites in plasma of rats in each group.

Ion Model	Retention Time (min)	M/Z	Component ID	Molecular Formula	Component Name	*P* Value
Positive	9.521	235.1691	HMDB00043	C_5_H_11_NO_2_	Betaine	2.48 × 10^−10^
	10.305	395.2206	HMDB04063	C_10_H_15_NO_3_	Metanephrine	1.21 × 10^−11^
	6.889	299.1313	HMDB06101	C_18_H_18_O_4_	Enterolactone	0.000245555
	2.431	252.1095	HMDB01983	C_10_H_13_N_5_O_3_	5′-Deoxyadenosine	0.0000987
	1.168	144.1020	HMDB04827	C_7_H_13_NO_2_	Proline betaine	0.011885662
	4.556	157.0608	HMDB02320	C_6_H_8_N_2_O_3_	Imidazolelactic acid	0.000000336
Negative	3.679	227.1416	HMDB11175	C_11_H_20_N_2_O_3_	L-leucyl-L-proline	1.32 × 10^−9^
	6.989	311.2237	HMDB10208	C_18_H_32_O_4_	15,16-DiHODE	0.000509626
	4.920	277.1455	HMDB01518	C_16_H_22_O_4_	Alpha-CEHC	0.00000215

**Table 2 molecules-24-02660-t002:** Enrichment analysis of metabolites.

Pathway Name	Total	Expected	Hits	Raw p	Holm p	FDR
Betaine Metabolism	10	0.109	1	0.105	1	1
Methionine Metabolism	24	0.262	1	0.235	1	1
Glycine, Serine and Threonine Metabolism	26	0.284	1	0.252	1	1
Tyrosine Metabolism	38	0.415	1	0.348	1	1

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
