# Peer review of "Dendrobium officinale Polysaccharides Inhibit 1-Methyl-2-Nitro-1-Nitrosoguanidine Induced Precancerous Lesions of Gastric Cancer in Rats through Regulating Wnt/β-Catenin Pathway and Altering Serum Endogenous Metabolites"

_molecules, 2019, doi:10.3390/molecules24142660_

Round 1

Reviewer 1 Report

This manuscript by Zhao reports that DOP suppresses the expression of Wnt/β-catenin pathway related genes and proteins to inhibit the progression of gastric cancer in rats. The manuscript also reveals that DOP regulates the endogenous metabolites in PLGC rats. The work appears to have not been reported before. The following items should be addressed prior to publication:

1. Ref 44 was a newly published paper on DOP protection against MNNG-induced PLGC from the same authors. It should be emphasized and discussed at the Introduction section. Summarized what was done in that paper and clarify what were the connections and differences between that published paper and this submitted manuscript.

2. The data provided in 2.1 Histological evaluation in stomach tissues of rats at the Result section were reported also in the Ref 44. The Fig. 1B and 1D in this manuscript were even identical to the Fig. 3B and 3D. I don’t think reporting a reported data at Results as a separated section is allowed. Please summarize the result from Ref 44 and cite the paper at introduction if the authors consider this part as important result need to be mentioned.  

3. The preparation of DOP model was not clear in the manuscript. Same experimental procedure was also described in Ref 44. Thus, please quote the language in Ref 44 and cite Ref 44 at the corresponding text.

4. The language needs to be double-checked for accuracy and readability. Several mistakes are highlighted in the manuscript PDF attached.

5. The full names of the abbreviations should be placed where first time the terms were discussed. Examples were provided in the PDF attached.  

6. At the Result section, the Figures were only cited at the end of each paragraph even when there are Fig. A, B, C .. in one figure. Please put FigX A, B or C right after the discussion at the text. Please see the PDF attached.  

7. at Line 243 at the manuscript, the author states that DOP could activate the Wnt/β-catenin pathway to inhibit the progression of gastric cancer in rats. Is that correct? DOP could suppress the expression of Wnt/β-catenin pathway related gene and protein, which means DOP inactivate the Wnt/ β-catenin pathway. Isn't it?

8. In the manuscript, there are 4 models (CTRL, PLGC model, H-DOP, M-DOP, and L-DOP). However, sometimes when the authors discuss about the results for different models, it was not clear which one they were talking about. See PDF attached, in which some examples are highlighted.

9. Please keep the format of References and Notes constant.

In addition to the comments above, some minor revisions required before publication are detailed at the PDF attached.

Reviewer 2 Report

Zhao et al report the use of Dendrobium officinale extract to inhibit PLGC-induced gastric carcinogenesis in rats. 

The manuscript presents interesting data, however major problems need to be addressed:

-      Major English review is necessary, with special attention to grammar and sentence construction. 

-      The major compounds present in the extract of Dendrobium officinale should be described

-      Restructure the introductions, it goes from cancer to metabolomics to cancer again.

-      Please redefine metabolomics

-      Define all the abbreviations used 

-      Figure 1, are all the images presented from the stomach, some look like intestine, please clarify

-      Figures 3 and 4, it is unclear what is been analyzed, serum or urine?

-      In general all the results presented lack the justification of why the experiment was preformed. Particularly: 

o  point 2.1: there is no explanation of thee model and the treatment strategy, had to go all the way to the methods to see it, and still not completely clear. Maybe a diagram will make the reading easier;

o  point 2.5, don’t understand why this experiment was done, is there a difference between groups?

o  Point 2.6, please clarify the differences;

o  Points 2.5 to 2.9, these experiments need more details on why they were performed and what are the major differences. Just stating that there are differences is not sufficient. This makes it really hard for the reader to understand.

-      The discussion is very confusing, since there are no partial conclusions anywhere or real discussion of the findings in the manuscript. There is a lot of references to work of others, but is lacking the explanation of how correlates with the results showed, and possible meaning of the findings. 

-      Several times there is discussion of c-Myc, but no results are show. Can you provide at least the western blot results for c-myc?

-      All western blot used for quantification of the animal samples should be shown in the supplementary materials

Reviewer 3 Report

The manuscript “ Dendrobium officinale polysaccharides inhibit MNNG induced-2 PLGC in rats though regulating Wnt/β-catenin pathway and 3 altering serum endogenous metabolites” is very interesting, complex, with well-described studies, the materials and methods are arranged in a systematic way and shows laborious discussions.

There are only a few minor mistakes such as:

- Layout of the manuscript

- Italic writing of the species

- The lack of spaces between brackets and some words

- The lack of a list with abbreviations because there are many in the text

-Line 377- specify the type of device used.

Round 2

Reviewer 1 Report

The authors revised the manuscript nicely according to the reviewers' comments. The English was also greatly improved. It is ready for the publication at the Molecules

Author Response

Thank  you very much for your hard working and give me some meaningful advice.

Reviewer 2 Report

The reviews made to the manuscriptDendrobium officinale polysaccharides inhibit 1-Methyl-2-nitro-1- nitrosoguanidine induced precancerous lesions of gastric cancer in rats through regulating Wnt/β-catenin pathway and altering serum endogenous metabolites have greatly improved the manuscript. However, some minor corrections are still warranted:

-      Line 67: Drug affection should be replaced by drug effects

-      The definition of Metabolomics does not refer in what the expected differences are

-      Line 74: very commonly is redundant, remove very

-      Review the use of a and an throughout the manuscript (for example an shouldn’t be used before words started with h)

-      Line 110: after the reference the sentence looks like it was cut, as it is it was no meaning 

-      Lines 203 and 227: is fine to put in the p values, but if you show the values you should also show the symbols. Optional you can say significant and in between parenthesis say “all p<0.05”

-      Line 225: Should read Figures 3A-D

-      Line 257: Figures 4A and B is repeated in the parenthesis

-      Line 403: should add references for the comments in colorectal, hepatocellular and gastric cancers 

-      Line 437: Again, references are missing.

Author Response

Point 1: Drug affection should be replaced by drug effects

Response 1: Thank you very much. I had reviewed it.

Point 2: The definition of Metabolomics does not refer in what the expected differences are

Response 2: Thank you for your advice, which will make the article more logical and easier to be understudied. I had revised.

Point 3: Line 74: very commonly is redundant, remove very

Response 3: Thank you very much. I had deleted “very”.

Point 4: Review the use of a and an throughout the manuscript (for example an shouldn’t be used before words started with h)

Response 4: Thank you for your advice. In the United States, the “h” in “herb” is silent. In Britain, it’s sounded. We say “an ’erb” while the British say “a herb” , so used “an” in front of “herb”.

Point 5: Line 110: after the reference the sentence looks like it was cut, as it is it was no meaning

Response 5: OK, I had checked them and made them into a sentence.

Point 6: Lines 203 and 227: is fine to put in the p values, but if you show the values you should also show the symbols. Optional you can say significant and in between parenthesis say “all p<0.05”

Response 6: OK, I had changed them “all p<0.05”.

Point 7:Line 225: Should read Figures 3A-D

Response 7:Thank you very much. I had changed it.

Point 8: Line 257: Figures 4A and B is repeated in the parenthesis

Response 8: Thank you very much. I had deleted one.

Point 9: Line 403: should add references for the comments in colorectal, hepatocellular and gastric cancers

Response 9: OK, I added it.

Point 10: Line 437: Again, references are missing.

Response 10: Thank you very much. Because the information of the two metabolites is form the HMDB website, I didn’t add it in the first. This time I searched some published papers and added them in the references.

Reviewer 3 Report

Line 108-the names of the species italic

Still there are some spaces between brackets and some words.

Please correct the references 5 and 9

Author Response

Point 1: Line 108-the names of the species italic

Response 1: Thank you. I checked and revised them.

Point 2: Still there are some spaces between brackets and some words.

Response 2: Thank you for your work. I checked and revised them.

Point 3: Please correct the references 5 and 9

Response 3: Ok, I had replaced capital letters with lowercase letters.